# A Study on Fire Retardant and Soundproof Properties of Stainless Steel EAF Reducing Slag Applied to Fiber Reinforced Cement Boards

**DOI:** 10.3390/ma16103841

**Published:** 2023-05-19

**Authors:** Chuan-Wen Chou, Hung-Ming Lin, Guan-Bang Chen, Fang-Hsien Wu, Chen-Yu Chen

**Affiliations:** 1Department of Architecture, National Cheng Kung University, Tainan City 701, Taiwan; gege39th@gmail.com; 2Sustainable Environment Research Laboratories, National Cheng Kung University, Tainan City 701, Taiwan; hmlin@mail.ncku.edu.tw; 3Research Center for Energy Technology and Strategy, Department of Aeronautics and Astronautics, National Cheng Kung University, Tainan City 701, Taiwan; gbchen@ncku.edu.tw; 4Research Center for Energy Technology and Strategy, Department of Mechanical Engineering, National Cheng Kung University, Tainan City 701, Taiwan; bluehusky759@gmail.com

**Keywords:** circular economy, EAF-reducing slags, high-value building materials, fire retardant, soundproofing

## Abstract

In recent years, cases of the improper utilization of steel furnace slag have been widely reported, resulting in a crisis of nowhere for recycled resources such as inorganic slag. The misplacement of resource materials that originally had sustainable-use value not only has a great impact on society and the environment but also greatly reduces industrial competitiveness. To solve the dilemma of steel furnace-slag reuse, it is critical to find solutions to the stabilization of steelmaking slag under the innovative thinking of the circular economy. In addition to enhancing the reuse value of recycled resources, the balance between economic development and environmental impact is also crucial. The high-performance building material could provide a solution based on a high-value market. With the development of society and the increasing requirements for quality of life, the requirements for the soundproof and fireproof performance of lightweight decorative panels common in cities have gradually become popular. Therefore, the high performance of fire retardant and soundproofing could be the main development focus of high-value building materials to ensure circular economic feasibility. This study continues the research results of the application of inorganic re-cycled engineering materials in recent years, and the application of electric-arc furnace (EAF)-reducing slag to the development of base materials for reinforced cement boards, in order to complete the development of high-value panels with fireproof and sound-insulation properties in line with the engineering characteristics of the boards. The research results showed the optimization of the proportions of the cement boards with EAF-reducing slag as a raw material. The proportions of EAF-reducing slag to fly ash at ratios of 70:30 and 60:40 all met the requirements of ISO 5660-1 Class I flame resistance; the sound transmission loss in the overall frequency band can reach more than 30 dB, which is higher by 3–8 dB or more than the same board with similar specifications (such as 12 mm gypsum board) in the present building-materials market The products could be developed into building partitions and ceiling decoration boards with high performance in terms of fire retardant and soundproofing values, and also reduce the use of natural raw materials by more than 35%. The results of this study could meet environmental compatibility targets and contribute towards greener buildings. This model of circular economics would achieve energy reduction, emissions reductions, and be environmentally friendly.

## 1. Introduction

The circular economy is an economic development model of continuous recycling and utilization, replacing the linear economic model of manufacturing-production-disposal in the past industrial revolution. This model of material flow forms a cycle of “resources, products, and renewable resources”. It would generate only a small amount of waste within a whole system, or even achieve the goal of zero waste.

However, the current resource cycle is encountering the following problems:Jurisdiction and responsibilities for waste reuse are not comprehensive;The quality of recycled resources/products needs to be further improved;The management and control of the resource-recovery market are out of order;The quality of basic environmental information needs to be further strengthened;Disposal and management of reused products.

At present, the resource cycle is not perfect in terms of management and technology. The types of waste that most urgently need effective solutions to include inorganic sludge (D-0902), steelmaking furnace-refining slag (converter slag, desulfurization slag, electric arc furnace-reducing slag), and incinerator bottom slag, etc. There are nearly 5.4 million tons of recycled inorganic sludge resources generated every year. This kind of waste has a large output and various properties. Most of these have the potential to expand, so their reuse is limited. They can only be used for low-strength backfill materials (CLSM) or landfill materials, and even flow into agricultural land and fish farm depressions [1].

The engineering properties of steelmaking-slag materials are better than natural sand and gravel, but steelmaking slag has stability problems, which makes the reuse rate generally low in actual engineering applications. The traditional treatment methods still cannot convert the free lime and magnesium oxide into calcium hydroxide and magnesium hydroxide, so expansion and deterioration of the concrete after hardening will occur [2]. Therefore, follow-up stabilization is very important and a key factor in slag reuse requirements. Therefore, for the stabilization of steelmaking slag, in addition to the high-temperature steam curing method—the technology to solve the stabilization of steelmaking slag with low energy consumption—will be a critical factor in improving the reuse of steelmaking slag. In recent years, the production technology of the stainless-steel industry has continued to innovate and develop, and it is now a mature industry. As the output increases year by year, it will inevitably be accompanied by the production of process waste slag. Due to the limited land area of Taiwan, the disposal of slag by traditional burial methods will inevitably face the problem of disposal costs. How to effectively reuse it as construction and engineering materials is an important issue at present.

The high-value product of decoration boards constructed from EAF-reducing slag developed in this study has a wide range of applications. As building materials, cement boards can be applied to indoor partitions, ceilings, etc. Practical applications for the materials could be included in commercial buildings, department stores, restaurants, and residential buildings. There are many potential variations of the materials, and these can be widely used for other special purposes, such as wrapping beams, wrapping columns, decoration, exterior wall decoration, etc. The decoration boards, like fiber-reinforced cement boards, have the advantages of economies of scale for mass production, and can be produced in a standardized or automated manner. Due to the streamlined production process and the reuse of raw materials, it is necessary to coordinate with complete warehousing and logistics to make it a technology-intensive and capital-intensive industry. The high-value products of EAF-reducing slag promoted by this project—fiber-reinforced cement board, through the integration of interdisciplinary technology and the use of diverse materials, can adjust expenses and reduce costs. It can also effectively avoid the risk of a single material source being cut off, while integrating the industry supply chain from the material end to the end user and the market, launching high-value products that are functional, recyclable, low-cost, and comply with regulatory requirements.

The main material components of fiber-reinforced cement board are cement and inorganic mixed materials. From the perspective of the characteristics of the slag, there are two ways to carry out the fiber–cement board trial production. One is to directly replace the inorganic mixed material as the filling material of the board; the other is to replace the amount of cement. The main components of the reduced slag are CaO and SiO_2_. The potential of the component as raw materials in cement boards is used to achieve the original role of cement in bonding other materials [3].

The overall goal of this study was to develop building-material development technology for the stabilization and recycling of EAF-reducing slag. This study was mainly conducted to integrate the relevant raw material screening mechanism, manage the quality of raw materials for the cooperative manufacturer (Walsin Lihwa Co., Ltd., Tainan, Taiwan), and combine controllable density material technology, green building material evaluation, and verification technology to produce fire retardant and soundproof materials. These would then become high-quality green building materials.

The study focused on the fiber-reinforced cement board as the product of EAF-reducing slag reuse. The study of the product development process was carried out in two parts. The first was conducted to establish the screening mechanism of inorganic material characteristics and high-value management procedures. The second part was conducted to match the verification standards of environmental compatibility and slag-modification technology. The flow chart of the study is shown in Figure 1. The study presents a feasible solution for a low-energy consumption silicon-based modification method to solve the expansion problem of EAF-reducing slag, and the production of high-performance products for building materials based on the EAF-reducing slag.

## 2. Materials and Methods

### 2.1. Slag Controllable Density and Stabilization Technology

As the hydraulic characteristics and expansion characteristics of electric arc furnace slag gradually become clear, the utilization rate has also increased to 55% [4]. For the expansion problem of EAF slag, in addition to the content of f-CaO and f-MgO, the distribution of f-CaO and f-MgO in the steel slag and the change of the generated Ca(OH)_2_ have an impact on the volume stability also has an important impact [5]. Since f-CaO and f-MgO are basically locally concentrated or aggregated in the steel slag. The generated Ca(OH)_2_ and Mg(OH)_2_ are also locally concentrated, and the initially formed Ca(OH)_2_, Mg(OH)_2_ is mostly in the form of amorphous or small crystals. As the hydration progresses, the Ca(OH)_2_ and Mg(OH)_2_ crystals formed at the initial stage gradually increase in size and continuously squeeze the surrounding hydration products. Hardness would occur, and local expansion pressure and structural inhomogeneity generated inside the cement paste, which eventually would lead to serious consequences of expansion and cracking of structural materials [6,7]. According to the quality demands of application of EAF-reducing slag, details of the experiments are listed in Table 1.

EAF-reducing slag is a by-product produced in the production process of stainless steel. It is the same as most slag materials. Because it contains substances made unstable and harmful to influence of engineering quality such as free lime (f-CaO) [8], it is easy to generate volume during use. Expansion, which limits the engineering application surface, thus reducing or inhibiting its volume expansion, is the most important problem to be overcome in the engineering of slag materials [9,10,11].

The ratio design was divided into two types of product design: pouring products and high-pressure products. This time, we will mainly discuss two groups of formulas with ash-soil ratio of 0.35 and 0.45. To solve the problem of reducing the expansion of ballast, we mainly used silicon additives for modification, and the addition amount is divided into four ratios: 0%, 20%, 30% and 40%. The proportion of the design is shown in Table 2. The ratio design of high-pressure products was mainly based on the strength of the pouring products, and adding different pulp fibers to produce cement fiber boards with a density of 1.2 and 1.5, in order to discuss the mechanical properties and provide finished products for the sound insulation of building materials, as well as an evaluation of performance and thermal characteristics.

### 2.2. Visual Fire Retardant Technology

Ensuring fire resistance is a crucial property of building materials, especially when using circular materials in high-value building applications, which requires verifying their heat performance and fire resistance during materials proportion testing, following the relevant standards in Table 3. The combustion test method for building interior decoration materials will be used for the fire resistance test on all levels, considering the engineering application characteristics of various cement boards and complying with relevant national standards. To this end, a self-designed and constructed combustion detection system was employed to conduct the necessary fire-resistance experiments.

To test the fire-resistance of materials, it is necessary to use larger-sized test materials (20 cm × 20 cm). To facilitate this, a miniaturized fire-resistant material test furnace system was created using stainless steel 316, as shown as Figure 2. The furnace features an infrared ceramic furnace at the bottom for heat source. The top of the furnace accommodates cubes measuring 10 cm × 5 cm or plates with a thickness of 2 cm. One air inlet is used to provide air, and the upper four air outlets are used for exhaust gas collection and analysis. The furnace body’s upper cover has a high-temperature quartz plate observation window, which allows the entire testing process to be recorded and measured using an infrared thermal imager. The specimen part of the test uses a ceramic fiber board that is cropped to fit snugly around the specimen. The test follows the CNS 6532 and CNS 14705 flame resistance test specifications for building materials and decoration materials. Flame retardant materials must be heated continuously for 5, 10, and 20 min to be considered as flame-retardant grades 3, 2, and 1. Test platforms were selected from manufacturers that met the testing requirements, and long-term heating can be provided if necessary.

### 2.3. Small Specimens Sound Transmission Loss Measurement

In 2016, revisions were made to the Architectural Technical Regulations which included the addition of air-sound and sound-insulation performance evaluation standards for partition walls in the Soundproofing section of the Architectural Design and Construction Code, reflecting the significance of sound insulation in modern urban architecture and emphasizing the need for building materials that consider both circular economy and high performance in line with current domestic industry and policy development trends.

When it comes to testing the sound insulation of building materials, a full-scale and complete construction is required, which is challenging to achieve in the early stages of materials development. However, to evaluate the soundproof properties of specimens in different materials proportions and verify them alongside mechanical and heat performance, an impedance cube system was utilized in this study. The system consists of a cube with a diameter of 10 cm and a loudspeaker that emits sound in the range of 97–100 dB. Measurements were taken at both ends of the cube using a 1/4″ microphone and then analyzed using a four-channel analyzer to obtain the transmission-loss value measurement of the normal incident sound energy. The measurement range was between 80 and 1600 Hz, and this system provided an efficient way to adjust material proportions and assess soundproof properties in real time. The system is shown in Figure 3.

To avoid any sound leakage that could affect the measurement value, the test body, prepared as mentioned earlier, was placed at the center of the tube wall, and any gaps around it were filled. The test was conducted in two separate procedures, with the frequency ranges of 80~500 Hz and 400~1600 Hz. To measure different frequency bands, the distance between S1 and S2 was adjusted accordingly. After synthesizing the numerical values obtained from the different frequency bands, the overall transmission-loss performance within the range of 80~1600 Hz was determined. The content of the system is shown in Figure 4.

The study conducted a sound transmission-loss test on different materials, which were prepared using raw materials and poured test materials with a thickness of 20 mm. EAF-reducing slag was used as the raw material, which was crushed, screened, and mixed before pouring the materials into a round mold through a modification procedure. After 28 days of curing, the preparation was completed, and its high specific gravity made it suitable for subsequent development of the cement board with better soundproofing. The control part made of natural cement was set at the same proportion as the specimen, as specified in the table that shows the relevant specifications and scope of use of each specimen. The test included three types of specimens, each with different specifications as detailed in Table 2 and Figure 4 which were evaluated for their transmission-loss performance.

## 3. Results

From the results of the evaluation for EAF-reducing slags as building decoration boards, the characteristics would compare to the demands of performances from standards CNS 13777. The performances would include dry unit weights, bending strength, water absorption length change rate, flame resistance class, and sound transmission loss. According to the comparison of specimens, the basic mechanic characteristics have been shown higher strength than the same types of cement boards. With regard to environmental compatibility, the results of the toxicity characteristic leaching procedure were made available to the application of building materials. In fire retardant and soundproof properties, the results were also shown to be 1.2 to 1.5 times the values in performance compared to normal cement boards. An advanced description for several parts is presented below.

### 3.1. Application of Inorganic Polymerization Technology to Controllable Density Materials

On the basic mechanical properties of pouring products, the study mainly discussing the mechanical properties of cementing materials at 28 days. The measurement results would be the reference for the proportion adjustment of the second part of high-pressure building material products. The second part of the high-pressure product part mainly establishes the proportion design and mechanical test of the two groups of concrete fiber boards with a bulk density of 1.1 and a density of 1.3. The high-pressure production method of specimens would relate to the actual process of fiber-reinforced cement boards. In this part, the first part would show the results from the 28-day mechanical test of the pouring products as Table 4. The second part would show the results from the mechanical tests related to the specific gravity of water absorption, water absorption expansion rate, and bending test of high-pressure products. The results are described below as Table 5 and Table 6.

For the part of high-pressure products, mainly based on the results of the 28-day compressive strength of the irrigation products, two groups of C/S ratios of 0.35 and 0.45 were selected, with one proportion of 70% reducing slag and 30% fly ash, and the other proportion of 60% reducing slag and 40% fly ash. The ratio was carried out to establish the proportioning design of two groups of fiber–cement boards with dry unit weights of 1.2 and 1.5. The specimen was initially produced by pressing molding and high-temperature steam-curing. The density control can reach the design of two groups of ratios of 1.2 and 1.5. For the mechanical test part of high-pressure products, bending strength tests were mainly carried out according to CNS 13777, and the test results are shown in Table 6. The results show that when the S/F ratio is 1:1, the dry unit weight is 1.33–1.40 g/cm^3^ after drying at 60 °C for 24 h. According to the results of the bending strength test, as the ratio of C/S is 0.35, the proportion of 60% EAF-reducing slag and 40% fly ash shows the best strength, which is 9.97 N/mm^2^. The second is 9.39 N/mm^2^ with C/S ratio of 0.35, the proportion of 70% EAF-reducing slag and 30% fly ash. Furthermore, when S/F is 1:2, the dry unit weight is 1.11–1.15 g/cm^3^ when dried at 60 °C for 24 h. With the C/S ratio is 0.45, the results of the bending test showed that the proportion of 70% EAF-reducing slag and 30% fly ash is the best strength, which is 5.97 N/mm^2^. The second is 5.66 N/mm^2^ with C/S ratio of 0.35, the proportion of 70% EAF-reducing slag and 30% fly ash.

For the water absorption, dry unit weights, and water absorption length change rate tests of high-pressure products (fiber–cement boards), the tests are shown in Table 6 and Table 7. According to the requirements of CNS13777 cement board, when the dry unit weight is 1.1 g/cm^3^, its water absorption rate and water-absorption length change rate are not required in the specification [12].

The dry unit weight is 1.3 g/cm^3^, the water absorption rate is below 33%, and there is no requirement for the change rate of water absorption length. As for the dry unit weight of 1.5 g/cm^3^, the water absorption rate is below 28%, and the change rate of water absorption length needs to be 0.25%. The results show that in the group with S/F as 1:1, the dry unit weights are 1.28–1.32 g/cm^3^, the water absorption rate is 32.43–34.18%, and the change rate of water absorption length of the test body is 0.157–0.181%. As for the weight of S/F as 1:2, the dry unit weights are 1.01–1.14 g/cm^3^, the water absorption rate is 40.91–49.67%, and the change rate of water absorption length of the test body is 0.158–0.184%. According to the results of the proportion design of this project, the proportion of dry unit weights at 1.3 g/cm^3^, and its water absorption is just around the standard value. The rate is lower than 0.25%.

### 3.2. Environmental Compatibility Verification

The total element analysis is mainly based on the standard method NIEA M301.00B & NIEA M104.02C of the Environmental Inspection Institute. Microwave digestion equipment and an inductively coupled plasma emission spectrometer were used for testing, and an X-ray fluorescence spectrometer (XRF) was used to assist in confirming the correctness of the data. The EAF-reducing slag raw materials were subjected to microwave-assisted acid digestion, and the concentration of cations in the solution was analyzed by an inductively coupled plasma emission spectrometer, and the oxide content was converted to the analysis results of the X-ray fluorescence spectrometer (XRF) listed in Table 8 and Table 9.

As shown by the analysis results, the main components are CaO, MgO, SiO_2_, Al_2_O_3_, etc. The content of other main components such as Fe_2_O_3_, MnO_2_, TiO_2_, etc. is relatively low. Overall, the proportions of elements in the main components presented by the two analysis methods are similar. In the secondary composition, XRF found that the content of Cr_2_O_3_ was higher than the results of microwave digestion and ICP-OES analysis, which may be due to the difference in the sensitivity of the instrument to specific elements.

EAF-reducing slag was analyzed for heavy metal concentration after the standard method of the Environmental Inspection Agency—Toxic Characteristic Dissolution Procedure (R201.15C) [13]. The results are shown in Table 10. It can be seen from the table that the results of the dissolution concentration of heavy metals in stainless steel reducing slag are all lower than the TCLP dissolution standard value in the hazardous industrial waste identification standard.

To confirm the safety of the EAF-reducing slag used in watering products, the heavy metals concentrations of the finished product produced by using the EAF-reducing slag as a raw material were analyzed by the standard method of the Environmental Inspection Administration-toxicity characteristic dissolution procedure (R201.15C). The results are shown in Table 11. It can be seen from the table that most of the heavy metals were below the detection limit in the samples containing the proportion of the reduced ballast, and only the heavy metals barium, chromium, and selenium were eluted, but the eluted concentrations all met the current national standards.

### 3.3. Fire Retardant Properties of EAF Reducing Slags Used in Building Decoration Boards

Based on the formula design of fiber–cement boards reused by EAF-reducing slag in this study, the C/S ratio was set at 0.35 and 0.45, the design strength is 280 kgf/cm^2^ and 350 kgf/cm^2^, and the production methods included the pouring method (cement boards of EAF-reducing slag base without fiber) and high-pressure method (fiber-reinforced cement board of EAF-reducing slag). The high-pressure method was divided into density ≈ 1.3 and 1.1, and its thermal conductivity (k) value is in the following order: pouring method > high-pressure method (density ≈ 1.3) > high-pressure method (density ≈ 1.1) (as shown in Figure 5). Under the four ratios of C/S ratio as 0.45, the trend of the thermal conductivity (k) value is similar to that of the series of C/S ratio as 0.35 series, and slightly higher than that of the series of C/S ratio as 0.35 (see Figure 5). Above all, the fiber-reinforced cement board of EAF-reducing slag (density ≈ 1.1) had good thermal insulation properties when the C/S ratio is 0.35 and 0.45.

Figure 6, Figure 7 and Figure 8 show the results of the flame resistance test, demonstrating that the EAF-reducing slag foundation fiber-free cement board, the control group (C100) series test pieces of the pouring method, and the EAF-reducing slag fiber–cement board of the pressing method have densities ≈1.3, 1.1, the control group (C100 1:1), and (C100 1:2) series test pieces. Before the heat test, the board must go undergo pre-treatment procedures such as drying procedures [14]. After drying, the surface of the specimen has a nearly off-white cement surface with no cracks on the surface. After being subjected to the heat test, a large area of the surface of the heated surface exposed to the flame appears dark gray, while the back surface of the heated surface has no burning, penetration, cracks, or holes on the surface. The surface condition of the specimens with the pouring method, the EAF-reducing slag with fiber-free cement board and the control group (C100) series test pieces showed the same off-white color after drying. The cement board prepared with the high-pressure method, the density ≈1.3, 1.1, and the control group (C100 1:1 and C100 1:2) series of test pieces are gray-black, and the wall around the test piece is black-brown, and dark gray. The layer is caused by the gradient of the ceramic fiber board used to fix the test piece to black-brown and dark gray after being heated, and it is not caused by the characteristics of the test piece itself. The temperature-change image of the test piece taken by the infrared thermal imager and the whole heating test process were taken with a high-quality camera.

This study was based on the thermal characteristics, compressive strength, and bending strength, including tests on samples made using the pouring method (cement boards of the EAF-reducing slag base without fiber) and high-pressure method (fiber-reinforced cement board of EAF-reducing slag). The high-pressure method was divided into density ≈ 1.3 and 1.1, from which seven formulas were selected, including pouring method CS0.35-70-30, pouring method CS0.35-80-20, pouring method CS0.45-70-30, and high-pressure method CS0.35-70-30-1:1, high-pressure method CS0.45-70-30-1:1, high-pressure method CS0.35-70-30-1:2, and high-pressure method CS0.45-70-30-1:2 for the cone calorimeter test. As a result, it passed the flame-resistance level 1 test, which will help to replace natural materials in the future and reduce manufacturing costs [15]. According to the test results of the cone calorimeter, the high-pressure method CS0.35-70-30-1:2 and the high-pressure method CS0.45-70-30-1:2 in the flame resistance level 1 test all exceed the total heat release of the first point of the item: the total heat release of flame-resistant grade 1 materials was below 8 MJ/m^2^, but in the second point, the total heat release is below 15 MJ/m^2^, and the b parameter calculated according to A.2 of CNS14705-3 was below −0.4 [16], confirming that the calculation results meet the first level of flame resistance.

### 3.4. Soundproof Properties of EAF-Reducing Slags Used in Building Materials

According to the results of the transmission loss of each specimen, as shown in Figure 9 and Table 12, the effects provided by the boards of the building materials are mainly rigidity and strength. Therefore, reflecting the influence of its specific gravity and thickness on the transmission-loss performance, the middle and low frequencies are mostly the main contribution range of the board’s sound-insulation performance. Since most of the fiber-reinforced cement boards used in this study are used for the main body of the boards for building construction, 80~1600 Hz was used as the comparison range of transmission-loss performance. In addition, based on the above-mentioned mechanical test results, the test was carried out with the optimal proportion the performance requirements of fiber-reinforced cement board, and the blending ratio of the fiber and fly ash was adjusted to confirm its benefits. The following section analyzes the transmission-loss characteristic values of various specimens and uses 12 mm gypsum board as a control group to make a comprehensive comparison of each material to confirm the benefits of different application methods.

The transmission-loss characteristics of the heavy board prepared with EAF-reducing slag and fly ash have good benefits in the low-frequency range, showing the advantage of higher density. In this ratio, the original low-frequency disturbance is effectively eliminated, its overall performance can reach more than 30 dB, and it can only reach more than 35 dB above 1000 Hz. The existing 10 mm thick board has the characteristics of high transmission loss in the middle and low frequencies. In the future, if it is further combined with the internal filling material matching the partition wall system, it will be feasible to subsequently develop heavy-duty sound insulation systems.

With the addition of 1:1 fiber content, the mechanical properties of the board can be improved to meet the requirements of standard specifications. It also has a good effect on the transmission-loss characteristics of low and medium frequencies. Under the influence of a slight decrease in specific gravity, it is difficult to increase the thickness above 500 Hz. However, at a density of 1.3, it can still achieve an average sound transmission loss of more than 30 dB. Compared with the same grade of 12 mm gypsum board, it can achieve more than 10 dB sound insulation advantage. In addition, under the existing frame-type compartment system, the board can be combined with different filling materials and board-assembly methods to improve the sound insulation effect for medium and high frequencies.

By increasing the amount of regenerated fiber to achieve light weight and maintain the mechanical strength of the heavy board, the overall frequency band value is reduced by 1–3 dB due to the impact of light weight. However, it can still be maintained above 20 dB, and it still has excellent transmission-loss characteristics below 500 Hz. Above 500 Hz, it is maintained below 30 dB, and it is difficult to increase it further. Same as the existing frame-type compartment system, it can be combined with different filling materials and boards assembly methods to improve the sound insulation effect of medium and high frequencies.

Based on the comparison of the above three groups, the feasibility of using slag as a fiber-reinforced cement board is confirmed by using the existing 12 mm gypsum board as the control group and comparing the transmission-loss performance. The board prepared with EAF-reducing slag and fly ash can be used as a heavy decoration board for subsequent applications. The performance of each frequency band is significantly higher than that of the control group. There is a difference of more than 10 dB in the middle and low frequencies. Still with 1–3 dB higher performance. The bending-resistant boards prepared with fiber material can correspond to the decorative boards for building construction in subsequent applications. There is still a difference of more than 8 dB in the performance of the middle- and low-frequency bands, and the high frequency has the same performance. Boards with improved fiber-reinforced flexural properties have a higher performance of 3–7 dB below 400 Hz, while the mid- and high-frequency parts are relatively insufficient.

## 4. Discussion

This plan was mainly developed with EAF-reducing slag as the material source. The target product was the fiber-reinforced cement boards used for partition wall and ceiling decoration. The developed high-pressure fiber–cement board underwent a process adjustment to improve the mechanical strength of the board, so that the product specification can be upgraded from CNS 3802 to the requirements of CNS 13777, and its performance can be compared with the general commercially available calcium silicate board.

Based on the doubts and risks in the use of heavy metals in the finished products of EAF-reducing slag, this study used different environmental conditions and contact times to explore the long-term release of harmful heavy metals in the environment, so as to fully evaluate the environment of resource recycling-product compatibility [17,18]. Through the environmental compatibility assessment and analysis, the building materials containing the reduced ballast were determined to not be impactful or harmful when used in the environment. The average values of heavy metals dissolution in different accumulation periods were all within the monitoring and control standard values. From this, it can be confirmed that the water quality of the water body contacting the building materials made of reduced ballast is non-polluting and harmful. The use of materials containing the reduced ballast in the environment can meet environmental safety standards [19,20].

For the thermal characteristics test, the thermal conductivity of these test pieces was monitored at length during the proportioning process, and it was found that the thermal conductivity (k) of the test pieces maintained a good consistency during the proportioning process, and maintained good heat-resistance and heat-insulation effects. According to the measurement results of thermal conductivity (k), under the test of CNS 14705-1 “Test Method for Combustion Heat Release Rate of Building Materials—Part 1: Cone Calorimeter Method”, the specimens with the ratio of EAF-reducing slag and fly ash for any series at 70%: 30% could all pass the first class of flame resistance.

Based on the improvement of soundproof performance, a total of seven groups of soundproof performance tests of the fiber–cement board samples were completed. It was confirmed by the test results that the transmission loss of its sound insulation performance is more than 3–8 dB better than that of the same grade board (12 mm gypsum board), and the optimal ratio can reach more than 30 dB in the overall frequency band.

Above the results of measurement, the performances meet the standards of CNS 13777 as shown in Table 13. The high-value products developed through this study and those meeting the standards account for at least 12.1 billion wall and ceiling applications of 37 billion annual interior decoration projects. It is expected that use of the materials will reduce the use of natural resources by about 35%, and at the same time reduce, by about 28–32%, the production cost of building materials. According to the existing national market demand for decoration boards, it is estimated that the decorative panels made by reusing waste can effectively reduce processing costs by nearly 500 million per year.

Comprehensive development results showed that for effective use of the established stainless-steel slag for cement products technology, the low-energy consumption silicon-based modification method solved the expansion problem of stainless-steel slag, while reducing the amount of cement and improving the bending strength of the specimens [21,22]. Combined with the testing technology for thermal characteristics, it was found that the modifier has a significant effect on improving sound insulation and heat insulation. The results are conducive to the products developed in this study meeting the mechanical strength, sound insulation performance, and thermal performance specifications required for market access.

## 5. Conclusions

In this study, EAF-reducing slag was developed as a raw material for fiber-reinforced cement boards and combined with an upgrading technology. The optimal ratio is 0.35 C/S ratio, 70% reduced slag, and 30% fly ash, with the S/F ratio being 1:1. Through the modification of CaO in the EAF-reducing slag and SiO_2_ in fly ash, the strength of the cement pouring board meets the standard, and can be increased by about 1.55 times. The admixture of EAF-reducing slag through high-temperature sintering can increase heat insulation by 25%. Furthermore, the soundproofing can be increased by 15% due to the greater density of the EAF-reducing slag to natural materials. The performance of the cement high-pressure board meets the requirements of CNS 13777, as shown in Table 13, and its performance is equivalent to the calcium silicate board on the market. Due to its relatively high ratio of replacement from EAF-reducing slag, the cost of production from building materials would be reduced by 28% to 32%. The results of the study will help to replace natural materials in the future, reduce manufacturing costs, and promote the trend of circular economics in the building industry.

## 6. Patents

This research was partially supported by the Ministry of Science and Technology of Republic of China under the Grant number MOST 108-2218-E-006-031. The EAF-reducing slags were totally supported by Walsin Lihwa Corporation.

## Figures and Tables

**Figure 1 materials-16-03841-f001:**
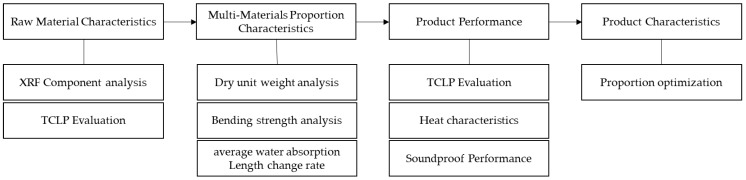
Flow chart of the study.

**Figure 2 materials-16-03841-f002:**
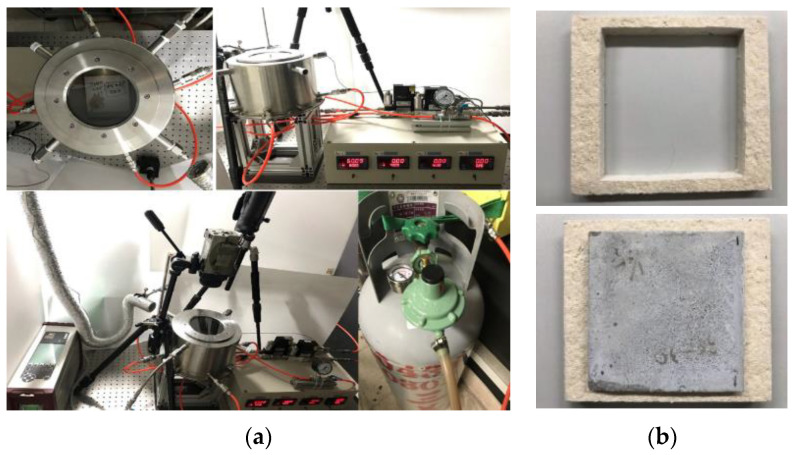
Miniaturized fire-retardant material test furnace system and specimens fixing frame: (**a**) system assembling; and (**b**) the fixing pattern of the specimen.

**Figure 3 materials-16-03841-f003:**
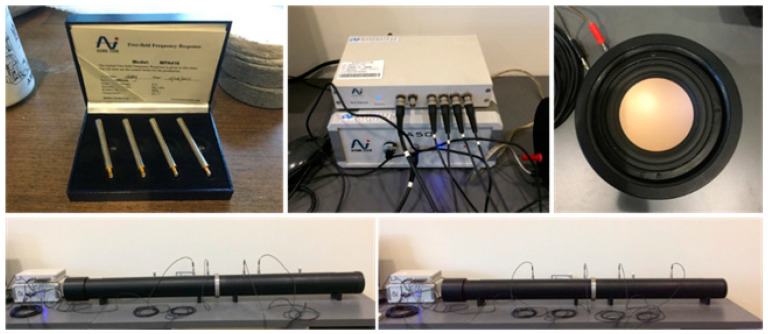
Impedance tube system and 1/4″ microphone.

**Figure 4 materials-16-03841-f004:**
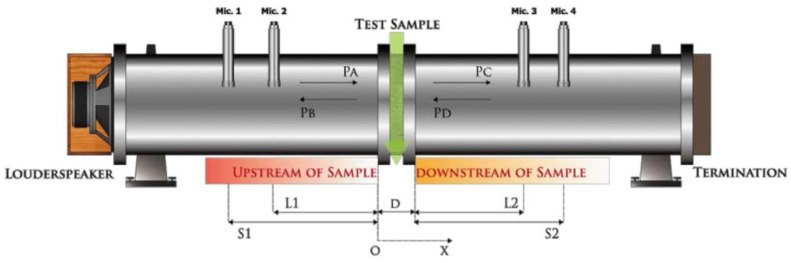
Content of sound transmission-loss measurement.

**Figure 5 materials-16-03841-f005:**
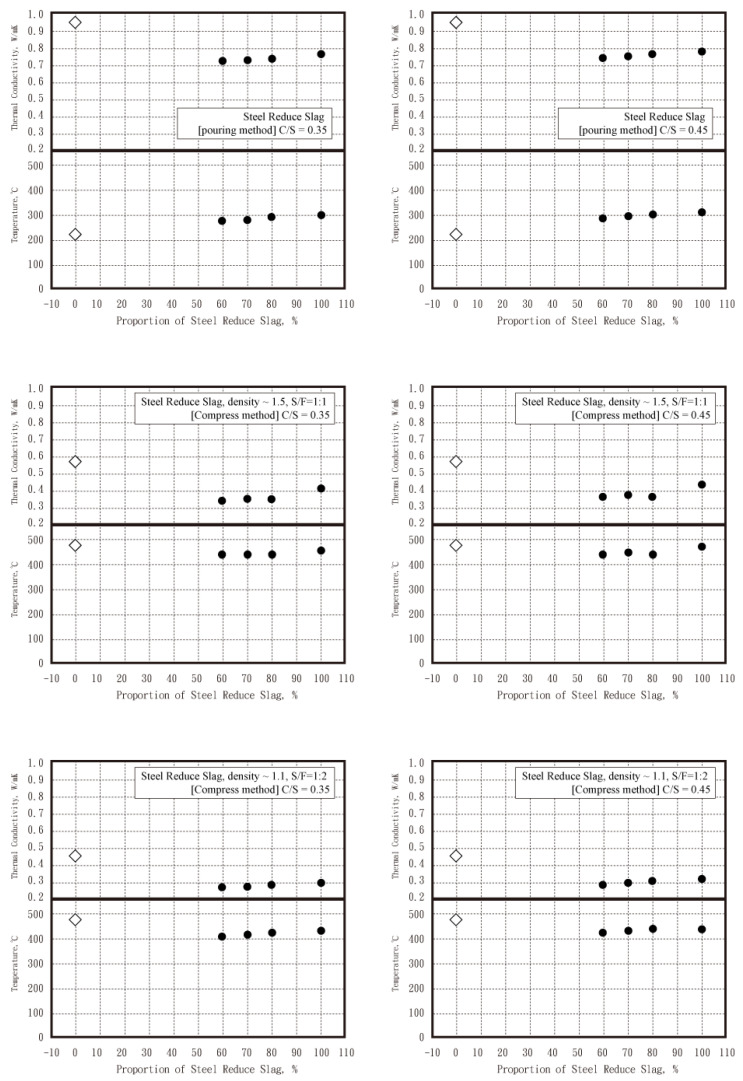
Temperature distribution and thermal conductivity of the EAF-reducing slag specimens in three settings of ratios. (white rhombus as specimens without EAF reducing slag, and black circle as specimens with EAF reducing slag).

**Figure 6 materials-16-03841-f006:**
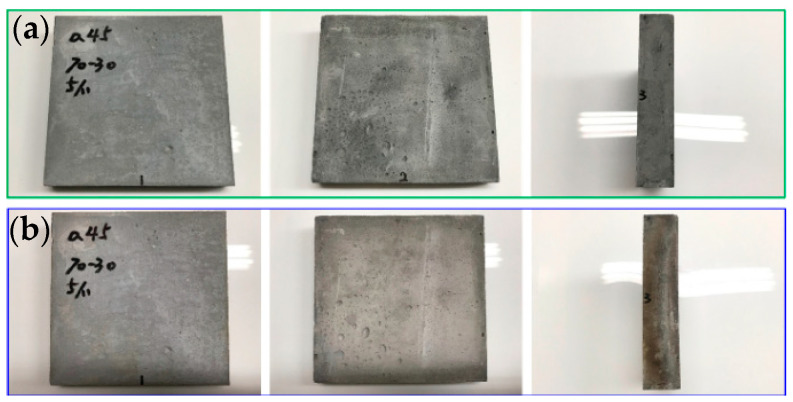
#40-70-30-C/S = 0.45: (**a**) before testing; and (**b**) after testing.

**Figure 7 materials-16-03841-f007:**
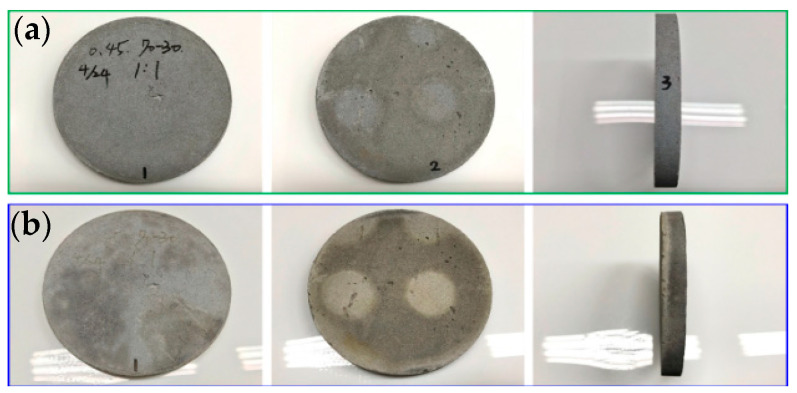
#40-70-30-C/S = 0.45 1/1: (**a**) before testing; and (**b**) after testing.

**Figure 8 materials-16-03841-f008:**
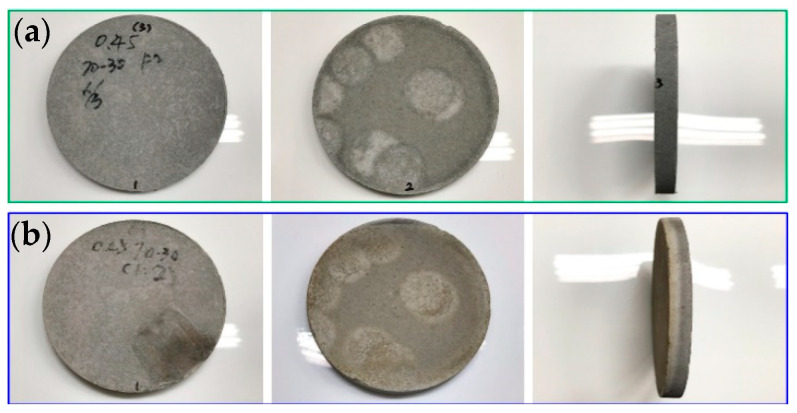
#40-70-30-C/S = 0.45 ½: (**a**) before testing; and (**b**) after testing.

**Figure 9 materials-16-03841-f009:**
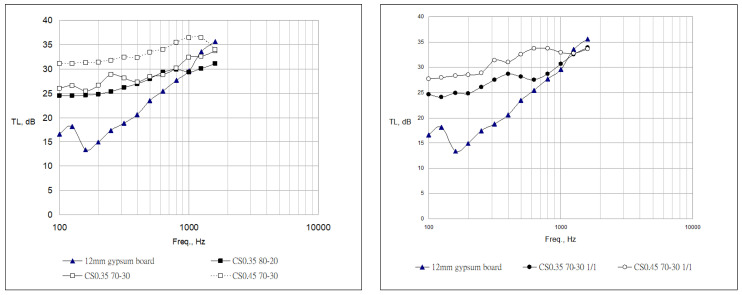
Sound transmission-loss curves of different combinations.

**Table 1 materials-16-03841-t001:** Mechanical performance examines.

Examine Subjects	Details and Standards
Mechanical and Physical properties	Specific density, Particle size distribution, XRF
Combination	1. C/S * = 0.35, 0.45 2. Reducing slag percentage 100%, 80%, 70%, 60% 3. Fly-Ash percentage 0%, 20%, 30%, 40% 4. S/F ** = 1:1, 1:2
Method of test for compressive strength of cylindrical concrete specimens.	CNS 1232
Method of test for bending and impact of building boards	CNS 3904
Method of Test for Autoclave Expansion of Portland Cement	CNS 1258
Fiber reinforced cement boards	CNS 13777
Toxicity Characteristic Leaching Procedure.	NIEA R201.14C

* C/S means the Cement-Sand Ratio. ** F/S means the solid matter (cement, reducing slag, fly-ash)–fibers Ratio.

**Table 2 materials-16-03841-t002:** EAF Steel reducing slag decoration materials combination with compressing method.

Solid Matter (Cement, Reducing Slag, Fly-Ash)-Fibers Ratio (S/F)	Cement-Sand Ratio (C/S)	EAF Reducing Slags Adjunction Percentage *	Fly Ash Adjunction Percentage **	Proportion Code
1:1 1:2	0.35	100%	0%	CS0.35 100-0
		80%	20%	CS0.35 80-20
		70%	30%	CS0.35 70-30
		60%	40%	CS0.35 60-40
1:1 1:2	0.45	100%	0%	CS0.45 100-0
		80%	20%	CS0.45 80-20
		70%	30%	CS0.45 70-30
		60%	40%	CS0.45 60-40

* The adjunction would be the percentage of the sand replaced by EAF-reducing slags. ** The adjunction would be the percentage of the EAF-reducing slags replaced by the fly ash.

**Table 3 materials-16-03841-t003:** Related standards of building materials properties.

Demands of Basic Properties of Board		Fire Retardant Verification	
Fiber reinforced cement sidings	CNS 11699	Method of test for incombustibility of interior finish material of buildings	CNS 6532
Fiber reinforced cement boards	CNS 13777		
Regenerated fiber–cement boards	CNS 14890	Method of test for heat release rate for building materials—Part 1: Cone calorimeter method	CNS 14705-1
Fiber cement boards	CNS 3802		

**Table 4 materials-16-03841-t004:** The compressive strength values of EAF-reducing slag boards.

Combination	Average Compressive Strength (kgf/cm^2^)		
	At Day 7	At Day 14	At Day 28
#40 CS0.35 30−0	131.38	173.68	220.84
#40 CS0.35 30−30	128.64	202.61	227.43
#40 CS0.35 30−40	125.72	209.16	260.99
#40 CS0.35 50−0	135.06	217.88	268.45
#40 CS0.35 50−30	137.36	166.86	210.98
#40 CS0.35 50−40	161.54	206.05	291.88
#40 CS0.25 30−0	158.70	208.25	280.16
#40 CS0.25 30−30	152.97	223.83	262.94
#40 CS0.25 30−40	131.38	173.68	220.84
#40 CS0.25 50−0	128.64	202.61	227.43
#40 CS0.25 50−30	125.72	209.16	260.99
#40 CS0.25 50−40	135.06	217.88	268.45

**Table 5 materials-16-03841-t005:** Bending Strength of the EAF-reducing slag boards with compressing method.

Combination	C/S	F/S	Dry Unit Weight (g/cm^3^)	Average Bending Strength (N/mm^2^)
#40 CS0.35 70-30 1/1	0.35	1:1	1.34	9.39
#40 CS0.35 60-40 1/1			1.33	9.97
#40 CS0.45 70-30 1/1	0.45		1.33	8.63
#40 CS0.45 60-40 1/1			1.40	8.82
#40 CS0.35 70-30 1/2	0.35	1:2	1.13	5.66
#40 CS0.35 60-40 1/2			1.11	4.13
#40 CS0.45 70-30 1/2	0.45		1.15	5.97
#40 CS0.45 60-40 1/2			1.13	5.64

**Table 6 materials-16-03841-t006:** Water absorption of the EAF-reducing slag boards with compressing method.

Combination	Apparent Specific Gravity (g/cm^3^)	Water Absorption (%)	Average Water Absorption Length Change Rate (%)
#40 CS0.35 70-30 1/1	1.28	34.18	0.166
#40 CS0.35 60-40 1/1	1.29	34.18	0.157
#40 CS0.45 70-30 1/1	1.32	32.43	0.181
#40 CS0.45 60-40 1/1	1.28	33.60	0.174
#40 CS0.35 70-30 1/2	1.01	49.67	0.184
#40 CS0.35 60-40 1/2	1.05	46.06	0.159
#40 CS0.45 70-30 1/2	1.14	40.91	0.158
#40 CS0.45 60-40 1/2	1.05	47.61	0.182
#40 CS0.35 70-30 1/1	1.28	34.18	0.166

**Table 7 materials-16-03841-t007:** Autoclave expansion results of the different cement–sand ratios.

At 3rd Days			At 28th Days		
Combination	Expansion Ratio (%)	Sample Description	Combination	Expansion Ratio (%)	Sample Description
#40 CS0.35 100-0	–	Burst	#40 CS0.35 100-0	–	Burst
#40 CS0.35 80-20	0.111	Exceeds	#40 CS0.35 80-20	0.094	Invariant
#40 CS0.35 70-30	0.056	Invariant	#40 CS0.35 70-30	0.037	Invariant
#40 CS0.35 60-40	0.030	Invariant	#40 CS0.35 60-40	0.015	Burst
#40 CS0.45 100-0	–	Burst	#40 CS0.45 100-0	–	Invariant
#40 CS0.45 80-20	–	Burst	#40 CS0.45 80-20	0.210	Exceeds
#40 CS0.45 70-30	0.072	Invariant	#40 CS0.45 70-30	0.044	Invariant
#40 CS0.45 60-40	0.052	Invariant	#40 CS0.45 60-40	0.036	Invariant

**Table 8 materials-16-03841-t008:** Characterization of fly-ash.

Elemental Composition	Al_2_O_3_	BaO	CaO	CO_3_O_4_	Cr_2_O_3_	CuO	Fe_2_O_3_	K_2_O	MgO	MnO	Na_2_O	NiO	P_2_O_5_	SO_3_	SiO_2_	TiO_2_
contents (%)	30.37	0.16	4.20	0.02%	0.02	0.01	5.24	1.01	1.20	0.05	0.65	0.02	1.00	1.16	53.61	1.29

**Table 9 materials-16-03841-t009:** Characterization of the EAF-reducing slag.

Elemental Composition	SiO_2_	Al_2_O_3_	Fe_2_O_3_	CaO	MgO	K_2_O	SO_3_	TiO_2_	MnO_2_
contents (%)	35.09	0.74	0.28	57.87	4.04	--	--	0.33	0.36

**Table 10 materials-16-03841-t010:** Toxicity characteristic leaching procedure of the EAF-reducing slag.

Element	As	Ba	Cd	Cr	Cr^6+^	Cu	Hg	Pb	Se	pH
Ppm(mg/L)	ND	0.18	ND	0.15	ND	ND	ND	0.01	ND	11.40
Standard (mg/L)	5.0	100.0	1.0	5.0	2.5	15.0	0.2	5.0	1.0	

**Table 11 materials-16-03841-t011:** Toxicity characteristic leaching procedure of the EAF-reducing slag boards.

Elemental Composition Unit: ppm (mg/L)	As	Ba	Cd	Cr	Cr^6+^	Cu	Hg	Pb	Se	pH
CS0.35 100-0	ND	1.954	ND	0.056	ND	ND	ND	ND	0.010	10.108
CS0.35 80-20	ND	2.313	ND	0.011	ND	ND	ND	ND	0.008	9.131
CS0.35 70-30	ND	2.938	ND	0.011	ND	ND	ND	ND	0.013	8.871
CS0.35 60-40	ND	3.400	ND	0.009	ND	ND	ND	ND	0.008	8.153
CS0.45 100-0	ND	1.636	ND	0.058	ND	ND	ND	ND	0.008	10.653
CS0.45 80-20	ND	2.313	ND	0.016	ND	ND	ND	ND	0.010	9.900
CS0.45 70-30	ND	2.667	ND	0.013	ND	ND	ND	ND	0.009	9.363
CS0.45 60-40	ND	3.153	ND	0.011	ND	ND	ND	ND	0.012	8.873
Standard (mg/L)	5.0	100.0	1.0	5.0	2.5	15.0	0.2	5.0	1.0	

**Table 12 materials-16-03841-t012:** Sound-transmission loss of different combinations.

Combination	Frequencies (Hz)—Sound Transmission Loss (dB)												
	100	125	160	200	250	315	400	500	630	800	1000	1250	1600
12 mm Gypsum board	16.60	18.20	13.40	14.90	17.40	18.80	20.60	23.50	25.40	27.70	29.60	33.50	35.60
CS0.35 80-20	24.49	24.52	24.64	24.81	25.36	26.21	26.89	27.94	29.34	29.91	29.30	30.07	31.10
CS0.35 70-30	25.95	26.52	25.45	26.65	28.82	28.10	27.24	28.44	28.79	30.20	32.35	32.54	33.71
CS0.45 70-30	31.08	31.10	31.27	31.39	31.74	32.41	32.26	33.44	33.95	35.47	36.41	36.39	33.98
CS0.35 70-30 1/1	24.59	24.08	24.90	24.82	26.09	27.52	28.64	28.14	27.49	28.67	30.66	32.58	33.88
CS0.45 70-30 1/1	27.70	27.99	28.28	28.45	28.83	31.34	30.97	32.51	33.75	33.72	32.94	32.76	33.59
CS0.35 70-30 1/2	22.92	23.08	23.30	23.63	24.19	26.27	28.48	27.36	28.59	32.12	32.81	31.83	28.48
CS0.45 70-30 1/2	27.38	27.32	27.44	27.65	27.87	27.87	27.83	28.00	29.07	29.11	30.91	32.86	31.63
#40 CS0.25 50-40	16.60	18.20	13.40	14.90	17.40	18.80	20.60	23.50	25.40	27.70	29.60	33.50	35.60

**Table 13 materials-16-03841-t013:** Comparison of CNS13777 and measurement results.

Measurement	Dry Unit Weight	Bending Strength	Average Water Absorption Length Change Rate	Fire Retardant Class	Thermal Conductivity
**Standards**	CNS 13777	CNS 3904	CNS 13777	CNS 14705-1	CNS 7332
**Unit**	g/cm^3^	N/mm^2^	%	–	W/m·K
**Standards**	0.90~1.50	>5.6	<0.18	class 1	<0.44
**Results**	1.11~1.34	5.64~9.97	0.157~0.174	Class 1	0.21~0.41

## Data Availability

The data presented in this study are openly available at https://wsts.nstc.gov.tw/STSWeb/Award/AwardMultiQuery.aspx, accessed on 1 August 2022.

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
