# Peer review of "A Study on Fire Retardant and Soundproof Properties of Stainless Steel EAF Reducing Slag Applied to Fiber Reinforced Cement Boards"

_materials, 2023, doi:10.3390/ma16103841_

Round 1

Reviewer 1 Report

While your research includes many experiments and works, from my part, I feel that the novelty and description of your study could be strengthened significantly. Additionally, the introduction, abstract, and discussion in the current MS lacked the necessary clarity and structure required for publication.

I would suggest that you extensively revise the introduction, abstract, and discussion sections to clearly convey the significance and relevance of your work, as well as the novelty of your findings.

Author Response

Point 1: While your research includes many experiments and works, from my part, I feel that the novelty and description of your study could be strengthened significantly. Additionally, the introduction, abstract, and discussion in the current MS lacked the necessary clarity and structure required for publication.

I would suggest that you extensively revise the introduction, abstract, and discussion sections to clearly convey the significance and relevance of your work, as well as the novelty of your findings.

Response 1:

Thanks for the suggestion of the manuscript. We have refined the introduction, abstract, and discussion sections. The significance and relevance would be more clearly stated than before.

Reviewer 2 Report

The conducted work “A study on fire retardant and soundproof properties of stainless steel EAF reducing slag applied to fiber reinforced cement boards” is good. However, following comments should be addressed to further improve the paper:

A. GENERAL COMMENTS ON PAPER

1.     Explicitly mention the novelty and research significance of current work in last paragraph of introduction section with emphasis on scientific soundness. Also, it would be great to add recent relevant literature review from 2021-2023 papers in introduction section as there is no paper cited from 2021-2023.

2.     Avoid paragraph of few (1-4) sentences throughout the manuscript, particularly in results and discussions sections, e.g. lines 38-42, 104-107, 272-276, 290-292, etc.

3.     Avoid long sentences throughout the manuscript, e.g. lines 38-41, etc.

4.     Line 334: heading numbering should be 3.3. Line 438: heading numbering should be 3.4.

5.     Results should be further elaborated with scientific reasoning.

6.     A separate brief section (explaining the relevance of this research for practical implementation) may be added before conclusion section. 

7.     English Language should be improved throughout the manuscript.

B. SPECIFIC COMMENTS FOR IMPROVING FOCUSSED RESEARCH

1.     Line 554: it would be better to mention the requirements of CNS 13777 in conclusions.

2.     Line 555: performance in which terms, authors are talking about?

3.     Line 556: which thermal properties would be improved?

4.     Line 557: how much cost reduction is expected?

Author Response

Point A-1: Explicitly mention the novelty and research significance of current work in last paragraph of introduction section with emphasis on scientific soundness. Also, it would be great to add recent relevant literature review from 2021-2023 papers in introduction section as there is no paper cited from 2021-2023.

Response A-1:

The references about the slag reused as engineering materials in recent year are inserted in the manuscript. The references are as below.

  • SURANENI, Prannoy, et al. ASTM C618 fly ash specification: Comparison with other specifications, shortcomings, and solutions. ACI Mater. J, 2021, 118: 157-167.
  • LIU, Yiliang, et al. Research progress on controlled low-strength materials: metallurgical waste slag as cementitious materials. Materials, 2022, 15.3: 727.
  • WANG, Her-Yung, et al. Evaluation of the engineering properties and durability of mortar produced using ground granulated blast-furnace slag and stainless steel reduced slag. Construction and Building Materials, 2021, 280: 122498.
  • WANG, Her-Yung, et al. Research on engineering properties of cement mortar adding stainless steel reduction slag and pozzolanic materials. Case Studies in Construction Materials, 2022, 16: e01144.

Point A-2: Avoid paragraph of few (1-4) sentences throughout the manuscript, particularly in results and discussions sections, e.g. lines 38-42, 104-107, 272-276, 290-292, etc.

Response A-2:

Thanks for the suggestion. The results and discussion sections have refined as review comments.

Point A-3: Avoid long sentences throughout the manuscript, e.g. lines 38-41, etc.

Response A-3:

Thanks for the suggestion. The results and discussion sections have refined as review comments. Because of the complicated corresponding comparison with measurement results and different proportion, we have strived as much as possible to avoid long sentences but there are still some necessary description difficult to avoid.

Point A-4: Line 334: heading numbering should be 3.3. Line 438: heading numbering should be 3.4.

Response A-4:

Thanks for suggestion of the manuscript. The parts have been refined.

Point A-5: Results should be further elaborated with scientific reasoning.

Response A-5:

Thanks for suggestion of the manuscript. The results section has been refined with more scientific reasoning.

Point A-6: A separate brief section (explaining the relevance of this research for practical implementation) may be added before the conclusion section.

Response A-6:

The practical implementation of the results of the study has been described in the additional section in the discussion.

Point A-7: English Language should be improved throughout the manuscript.

Response A-7:

Thanks for the suggestion of the manuscript. We have reviewed the whole manuscript and refined the content.

Point B-1: Line 554: it would be better to mention the requirements of CNS 13777 in the conclusions.

Response B-1:

Thanks for suggestion. The requirement and measurements result of CNS 13777 have been described in Table 13.

Point B-2: Line 555: performance in which terms, authors are talking about?

Response B-2:

The part tried to show the better thermal and soundproof properties through the admixture of EAF reducing slag. The conclusion has been refined as clear description and table of index of different performances,

Point B-3: Line 556: which thermal properties would be improved?

Response B-3:

The performances of heat insulation would be improved. The conclusion has been refined as clear description and table of index of different performances,

Point B-4: Line 557: how much cost reduction is expected?

Response B-4:

Due to its relatively high ratio of replacement from EAF reducing slag, the cost of production from building materials would reduce by 28% to 32%. The description has been added to the conclusion and discussion.

Reviewer 3 Report

1. The work does not respect the Template. Check the references.

2. The author should refine all chapter of the paper.

3. Introduction – please insert the references.

4. Review the chemical formulas.

5. Lines 131-133 - “Because it contains engineering hazardous substances such as free lime (f-CaO) [12], it is easy to generate volume during use.” –

Is (f-CaO) a dangerous substance?

6. Full characterization of the stainless steel.

7. Full characterization of the stainless steel furnace slag.

8. Line 158 - “Table 4. Table 3. Related standards of building materials properties.” - Renumber the tables. Table 5 is missing.

9. XRD analysis and SEM.

10. Insert a flow chart for your research.

11. What is the novelty of the paper?

12. What is the added value of the paper?

13. A comparative analysis between the results obtained and those reported in the specialized literature is necessary. Insert a Table. A scientific discussion is necessary.

14. Line 364 “high-quality camera (as shown in Figure 44).” – Figure 44?

15. “In this study, stainless steel furnace slag (reduced slag) was developed as the optimal 549 ratio of cement-sand slab admixture combined with upgrading technology. The optimal 550 ratio is 0.35 cement-sand ratio, 70% reduced slag, and 70% fly ash. The amount is 30%, 551

and the fiber ratio is 1:1.” Please explain.

16. References must be improved.

Author Response

Point 1: The work does not respect the Template. Check the references.

Response 1:

Thanks for the suggestion. The format of references has been refined.

Point 2: The author should refine all chapter of the paper.

Response 2:

Thanks for suggestion of the manuscript. We have reviewed the whole manuscript and refined the content.

Point 3: Introduction – please insert the references.

Response 3:

The references have been inserted in the sections of the manuscript.

Point 4: Review the chemical formulas.

Response 4:

The study applied the CaO and f-CaO from EAF reducing slag and SiO2 from fly ash to generate the hydration and pozzolanic reaction. The cement board would be set through these two reactions.

The chemical formula of hydration is below.

2(2CaO‧SiO2)4H2O3CaO‧2SiO2‧3H2O+Ca(OH)2

2(3CaO‧SiO2)6H2O3CaO‧2SiO2‧3H2O+3Ca(OH)2

After the Ca(OH)2 generated from the hydration and the f-CaO and H2O, the pozzolanic reaction would be activated and the chemical formula is below.

3Ca(OH)2+ 2SiO23CaO‧2SiO2‧3H2O

3Ca(OH)2+ 2Al2O33CaO‧2Al2O3‧3H2O

Through these two reactions, the C-S-H colloidal would be generated. This colloidal is above 55% of the whole volume of cement compound, and the strength of the boards would mainly depend on the composition.

Point 5: Lines 131-133 - “Because it contains engineering hazardous substances such as free lime (f-CaO) [12], it is easy to generate volume during use.” –Is (f-CaO) a dangerous substance?

Response 5:

Thanks for the suggestion. The sentences have been refined as lines 151-152.

Point 6: Full characterization of the stainless steel.

Response 6:

Electric arc furnace (EAF) stainless steel process uses electric energy to heat up to 1,500~1,600°C to melt scrap iron raw materials and added CaO. Then use oxygen blowing to remove the impure components in the molten steel, such as FeO, MnO, SiO2, Al2O3, and other oxides, and react with each other to form a stable mineral phase.

The specific gravity of this mineral phase is lighter than that of the molten iron in the reduction layer, and it floats on the upper layer to form "EAF oxidizing slag". The molten iron is then moved to the refining furnace (LF). A large amount of CaO is added to reduce and remove oxygen and sulfur components in molten iron, and thereby refine it into steel. The slag produced at this stage is called "EAF reducing slag".

Point 7: Full characterization of the stainless steel furnace slag.

Response 7:

Oxidizing slag has high iron content, hard texture, and high proportion. It is a dark brown lump with stable physical and chemical properties. Reducing slag contains less iron, more CaO, and MgO, and is gray-brown powder or block. Because it contains part of free lime, it will expand and easily disintegrate when it meets water. It must be matured to stabilize it. When placed, it is easy to pulverize due to volume change caused by crystal phase transfer. It should not be directly used as civil engineering materials alone.

After the slag goes through the stages of crushing, screening, and other procedures in the storage yard, and then stabilizes and matures, it can become a resource-based product.

The existing treatment methods still cannot digest the f-CaO and f-MgO into Ca(OH)2 and Mg(OH)2. As a result, the problem of expansion and deterioration after the concrete is hardened will occur. Therefore, the subsequent stabilization of slag is very important, and it is also the focus of current reuse requirements.

The composition of slag varies with the characteristics of the steel produced, different raw materials, different steelmaking methods, different production stages, different steel types, and different furnaces, etc.

However, the characteristics of steel ballast in various countries are basically similar, and there are slight differences mainly in the composition and production amount. The produced slag or reduced slag is mainly composed of residual iron or iron oxide, calcium oxide, silicon dioxide, aluminum oxide, magnesium oxide, and manganese oxide.

Point 8: Line 158 - “Table 4. Table 3. Related standards of building materials properties.” - Renumber the tables. Table 5 is missing.

Response 8:

The tables have been renumbered in the manuscript.

Point 9: XRD analysis and SEM

Response 9:

The results of EAF reducing slag of XRD and SEM in 1000x and 2000x are as below.

Point 10: Insert a flow chart for your research.

Response 10:

Thanks for suggestion. The flow chart has been inserted as figure 1 in introduction section.

Point 11: What is the novelty of the paper?

Response 11:

The novelty of the paper is that EAF reducing slag as business waste can be produced into high-usage amount building materials and increase diversified recycling benefits.

Point 12: What is the added value of the paper?

Response 12:

The added value of the paper is that the EAF reducing slag is used as a building decoration material, which has high heat resistance and high sound resistance, and increases the high value of building materials. This would be the model of circular economics with account of materials recycling and economics benefits.

Point 13: A comparative analysis between the results obtained and those reported in the specialized literature is necessary. Insert a Table. A scientific discussion is necessary.

Response 13:

Thanks for the suggestion. The refined section has been descripted in the additional section in the discussion, and also added the table 13 for the comparison with measurements results and standards.

Point 14: Line 364 “high-quality camera (as shown in Figure 44).” – Figure 44?

Response 14:

The figures have been renumbered in the manuscript.

Point 15: “In this study, stainless steel furnace slag (reduced slag) was developed as the optimal 549 ratio of cement-sand slab admixture combined with upgrading technology. The optimal 550 ratio is 0.35 cement-sand ratio, 70% reduced slag, and 70% fly ash. The amount is 30%, 551

and the fiber ratio is 1:1.” Please explain.

Response 15:

This section shows the optimization proportion of EAF reducing slag and fly ash for the feasible products with bending strength, fire retardant, and soundproof. The conclusion has been refined as more clear description.

Point 16: References must be improved.

Response 16:

Thanks for the suggestion. The format of references has been refined and inserted to the content of the manuscript.

Round 2

Reviewer 1 Report

It can be accepted in its current form.

Author Response

Dear reviewer,

Thanks for the acceptance of the responses. 

Reviewer 3 Report

Dear Authors,

The answers (points 11 and 12) should be correlated with the current state of the art.

Point 11: What is the novelty of the paper?

Response 11: The novelty of the paper is that EAF reducing slag as business waste can be produced into highusage amount building materials and increase diversified recycling benefits.

Point 12: What is the added value of the paper?

Response 12: The added value of the paper is that the EAF reducing slag is used as a building decoration material, which has high heat resistance and high sound resistance, and increases the high value of building materials. This would be the model of circular economics with account of materials recycling and economics benefits

Author Response

Dear reviewer,

Thanks for your suggestion. We have revised the responses of 11 and 12. The quantitative description of performances and economic value would be shown below and attachment.

Point 11: What is the novelty of the paper?

Response 11:

The novelty of the paper is that the EAF reducing slag is used as the critical material for the high performances of the fiber reinforced cement boards. As the comparison of the measurement results from the general cement boards, which is the sample of proportion as 100-0, and the ones made with EAF reducing slag, the performances of heat resistance and soundproofing are promoted. As the Figure 5., the thermal conductivity is down from about 0.5 to less than 0.4 or even 0.3. This shows the superior heat insulation of at least 25% to 40% from the cement boards made of EAF reducing slag than the ones without EAF reducing slag. As the Figure 9. and Table 12., the sound transmission loss is up to 25 or 30 dB at mid and low frequencies. This shows the EAF reducing slag would promote the sound insulation of building decoration boards by about 15% to 40%. These high performances of measurement results in comparison discover that the EAF reducing slag would be transferred from harmful engineering materials to the critical raw materials for the fire retardant and soundproof building materials.

Point 12: What is the added value of the paper?

Response 12:

This paper has shown the EAF reducing slag as business waste can be produced into high-usage amount building materials and increase diversified recycling benefits. From this transforming of uses of EAF reducing slag, the added value of this study would be displayed in three parts. First is the decrease of natural materials, like cement or soil, for the manufacture of cement boards as building decoration materials. As the proportion of this paper, like CS0.35/70-30, the soils would be replaced as EAF reducing slag and fly ash from natural materials. This reduces the use of natural materials by more than 15%. With the huge amount of building materials market, this would create about 450 to 500 million TWD as saving cost. Second, EAF reducing slag as business waste would take a big cost to complete the clearance, such like burying it in a specified area, especially in Taiwan as an island with limited land. Through the results of this paper, the EAF reducing slag would be applied as the critical materials of the fire retardant and soundproof building materials instead of the harmful business waste. The saving cost for clearance and burying would be up to 60 million TWD according to the outputting of EAF reducing slag in just one steel mill (Walsin Liwah Corp,). Finally, the high performance with fire retardant and soundproofing would raise the value of cement boards made with EAF reducing boards. According to the practice of public construction, cement boards with superior heat resistance and soundproofing would be the certified high-performance materials that could raise the selling price by about 10 %. This will be a powerful force to promote high-value circular economy.
